# Microstructure and Nanohardness of Ti-48Al-2Cr Alloy Solidified under High Pressure

**Duo Dong [1], Li Liu [2], Dongdong Zhu [1,\*], Yang Liu [2], Ye Wang [2] , Xiaohong Wang [1] and Zunjie Wei [3]**

[1] Key Laboratory of Air-Driven Equipment Technology of Zhejiang Province, Quzhou University, Quzhou 324000, China; dongduohit@163.com (D.D.); hitxiaohong_wang@hotmail.com (X.W.)
[2] School of Materials Science and Engineering, Harbin University of Science and Technology, Harbin 150001, China; li_liull@163.com (L.L.); lyang805@163.com (Y.L.); wangye1984@hrbust.edu.cn (Y.W.)
[3] School of Materials Science and Engineering, Harbin Institute of Technology, Harbin 150001, China; weizj@hit.edu.cn
\* Correspondence: zhudd8@163.com; Tel.: +86-570-8026634

**Abstract:** In this work, the Ti-48Al-2Cr alloy, solidified under different pressures and temperatures, was investigated in detail. The effect of high pressure on the microstructure and nanohardness of the Ti-48Al-2Cr alloy was investigated by X-ray diffraction, scanning electron microscopy, transmission electron microscopy, and a nanoindenter XP testing machine. The results showed that the B2 phase disappeared after high-pressure solidification. Compared with ambient pressure solidification, high pressure led to the increase of ($\alpha_2 + \gamma$) lamellar structure and the decrease of $\gamma$ phase. The nanohardness of the lamellar structure was discussed based on the microstructure observation. When solidified at 5 GPa/1873 K, the hardness rose to 5.54 GPa, an increase of 60.5% compared with that solidified at ambient pressure. However, the increased holding temperature of 1973 K made the dislocation density in the lamellar structures greatly decrease, and reduced the structure's hardness to 4.48 GPa.

**Keywords:** TiAl based alloy; microstructure; nanohardness; high pressure; solidification

## 1. Introduction

As high temperature structure materials, TiAl-based alloys show excellent advantages, such as low density [1], high specific strength [2], good corrosion resistance and creep resistance [3], high hardness and strength [4]. Due to such outstanding properties, TiAl-based alloys have been considered to be promising materials for application in aircraft, gas turbines and automotive engine [5,6]. However, the low ductility [7–9] greatly decreases the reliability of the microstructures and limits their application, a problem which has attracted the attention of many researchers [10,11]. Niu et al. [12] reported that after forging, the cast Ti-43Al-6Nb-1B alloy transformed from fully lamellar (FL) structures to duplex (DP) structures. The microstructure was refined and the mechanical properties were improved significantly. Many studies have been done to optimize the mechanical properties of TiAl-based alloys through the addition of a third element. In view of the β phase stabilizer addition, Cr is a notable element to improve ambient temperature, ductility, and mechanical strength of the TiAl-based alloys, which has been widely adopted by many researchers. Zhu et al. [13] indicated that the ductility of Ti-48Al-2Cr can be improved through the rapid solidification method. At the wheel speed of 30 m/s, the mechanical properties showed a great increase. Wang et al. [14] have found that after spark plasma sintering, the Vickers hardness of the TiAl alloy is increased by 15%.

Pressure is an important thermodynamic parameter of metallic materials which can influence the solidification process, especially under high pressure [15,16]. The influence of the alloying element and temperature during solidification processes has been widely studied. However, due to the limitations of traditional technology, the effect of pressure has usually been ignored [17]. High pressure solidification has gained much attention because of its special features, such as the refined microstructure and the decreasing diffusion coefficient [18–21], and this technique is an important and robust approach widely used in basic and applied science as well as industrial applications. High pressures and high temperatures have revolutionary impacts in modern society, ranging from industrial applications such as the fabrication of superconducting and superhard materials, to exploration of the properties of materials obtained under extreme conditions [22–25]. These characteristics are different from alloys that have solidified under ambient pressure. Wang et al. [26] confirmed that high pressure increases the eutectic point from 5.69% under ordinary pressure to 26.49% under 2 GPa, and the velocity of dilute solute solidified at interface declines exponentially. Wei et al. [27] reported the tensile strength and yield strength of Al-20Mg alloy increased to 467 MPa and 245 MPa respectively, because of solution strengthening and deformation strengthening caused by high pressure heat treatment. Nanoindentation is a widespread test technique for measuring the mechanical properties of the nanoscale, the microscale, and the thin-films materials [28,29]. It makes it possible to determine the hardness of the material, especially the hardness of target phase from hardness-depth data [29,30].

Up to now, the microstructure and mechanical properties of TiAl-based alloys solidified under high pressure has been rarely reported. Alloying elements such as Cr, Nb and Zr were added to improve the room temperature brittleness of the TiAl-based alloy [31]. Furthermore, it has been considered that 2% (at. %) Cr additions added to Ti-48Al alloy have the best room temperature ductility [13]. Therefore, the Ti-48Al-2Cr alloy was chosen to study the evolution of the microstructure and mechanical properties when solidified under high pressure in this work. The variation of the phase transformation and nanohardness of Ti-48Al-2Cr alloy has been studied.

## 2. Materials and Methods

A nominal composition of the Ti-48Al-2Cr (at. %) alloy was selected for melting by induction skull melt (ISM) under an Ar atmosphere. The main raw materials were pure sponge titanium (99.95%), aluminum ingot (99.99%), and high purity chromium (99.99%). The samples for high pressure solidification were cut into small cylinders with a diameter of 6 mm and a length of 9 mm by electrical discharge machine. The high pressure solidification experiments were conducted using the tungsten-carbide six-anvil apparatus (CS-IB), which can achieve the highest pressure up to 10 GPa and the temperature up to 2073 K. The pressure was determined by Bi's phase transformation [32]. The sample was heated in the graphite furnace in this equipment and the assembly of sample was shown in Figure 1. Because BN ceramic is a favorable heat-transmitting material which is stable for TiAl-based alloy melting, the sample was wrapped in BN ceramic to avoid the reaction between the graphite and high-pressure samples during the high pressure solidification process. Pyrophyllite was used as an encapsulant material, and can transmit pressure to the high-pressure samples. A thermocouple (W3%Re–W25%Re, type D) was used to measure the temperature of the samples. During high pressure solidification process, the pressure was first increased up to 5 GPa, and then the temperature was heated up to the target temperature, 1873 K and 1973 K, respectively. When the samples reached the preset pressure and temperature, the molten parameter was held for 30 min to obtain the fully melted samples. After that, the samples were cooled down to room temperature. The Ti-48Al-2Cr alloy prepared by ISM was also investigated for comparison.

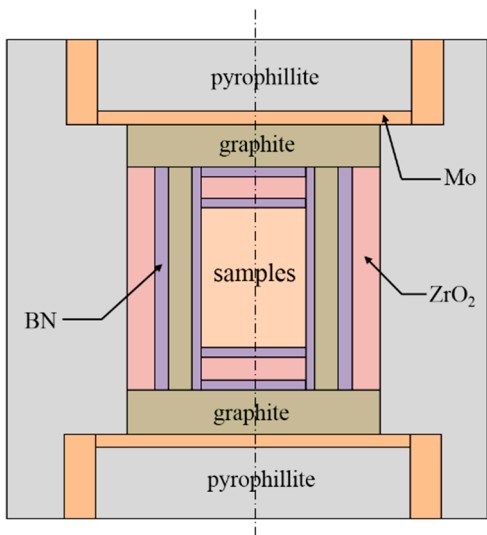

**Figure 1.** Schematic illustration of the assembly sample for high pressure solidification.

In order to study the microstructures and properties of samples, the surface was mechanically ground up to 2000 grit SiC paper, polished using diamond paste, and etched before microstructure observation. The phase composition of the samples was determined by X-ray diffraction (XRD) with Cu K$\alpha$ radiation, and the samples were scanned in a 2$\theta$ range of 15–90° angular region. The microstructure and chemical composition were characterized by scanning electron microscope (SEM) (SU8010, Hitachi) and energy dispersive spectroscopy (EDS). The samples were etched in Kroll's solution (5 vol.% $HNO_3$, 5 vol.% HF and 90 vol.% $H_2O$) and cleaned using an ultrasonic cleaning machine for 3 min in ethanol. The transmission electron microscopy (TEM) (Philips CM12) samples were prepared via twin-jet polishing. Here the twin-jet solution was mixture of 6% perchloric acid, 34% butanol and 60% methanol. The nanohardness of the samples was measured by the Nanoindenter XP testing machine (Nano Test, Vantage 7) (Aglient company, USA) at room temperature with the loading of 50 mN, and holding time of 10 s. The testing was performed 5 times on the target phase to obtain the average values.

## 3. Results and Discussion

### 3.1. The Phase Composition of the Ti-48Al-2Cr Alloy

Figure 2 shows the XRD patterns of the Ti-48Al-2Cr alloy solidified under ambient pressure at 1873 K, 5 GPa at 1873 K, and 5 GPa at 1973 K, respectively. Combined with the results of point analysis (EDS), it can be clearly known that the Ti-48Al-2Cr alloy was composed of a TiAl phase ($\gamma$) and a $Ti_3Al$ phase ($\alpha_2$) when solidified under different processes. High pressure solidification doesn't change the phase constitution of the Ti-48Al-2Cr alloy. However, the intensity of $\gamma$ phase decreases greatly as the solidification pressure increases to 5 GPa. In particular, when the solidified condition was 5 GPa and 1973 K, the diffraction peak intensity of $\gamma$ phase remarkably decreased and diffraction peak broadening also occurred, indicating that the grain size of the Ti-48Al-2Cr alloy decreases with the increasing solidification pressure.

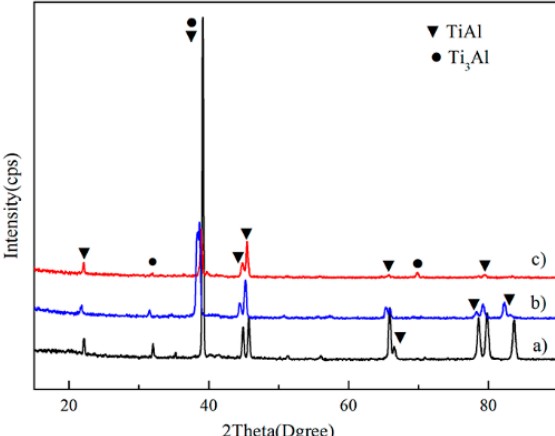

**Figure 2.** XRD patterns of the Ti-48Al-2Cr alloy solidified under different pressures. (**a**) ambient pressure, 1873 K; (**b**) 5 GPa, 1873 K; (**c**) 5 GPa, 1973 K.

### 3.2. Microstructure of the Ti-48Al-2Cr Alloy Solidified under Different Processes

According to the phase diagram of the Ti-Al-Cr ternary alloy [33], the solidification process of the Ti-48Al-2Cr alloy is Liquid (L) → β (primary phase), and L + β → α (peritectic transformation) occurs after continuous cooling. Then, the residual β phase transforms into the α phase and the B2 phase. During subsequent cooling, the lamellar structure is formed by $\alpha \rightarrow \alpha_{residual} + \gamma \rightarrow (\alpha_2 + \gamma)_{lamellar} + \gamma$ transformation ($\alpha_2$-Ti$_3$Al, γ-TiAl).

Figure 3 shows the microstructure of the Ti-48Al-2Cr alloy under different solidification conditions. As shown in Figure 3a, there are a large number of γ phases in the alloy after solidification at ambient pressure. When solidified at 5 GPa, the volume fraction of γ phase decreases while the lamellar structure increases (Figure 3c). Because of the shifted eutectic point to higher Al content and the increased eutectic reaction temperature, the content of the lamellar structure increases, according to the lever rule [34–36]. The increase of superheat will not only make the element distribution more uniform, but will also improve the thermodynamic undercooling and reduce the content of interdendritic γ phase (Figure 3d) [22]. However, according to Gao's study [37], the effect of temperature on the size and shape stability of microstructural features has been extensively investigated, the lamellar structure would be coarsened with increasing superheat. When the solidification condition of the Ti-48Al-2Cr alloy changed from ambient pressure to 5 GPa at 1873 K and 5 GPa at 1973 K, the volume fraction of γ phase decreased from 43% to 29.4% and 11.2%, respectively (Figure 3a,c,d).

Few B2 phase particles can also be found in the Ti-48Al-2Cr alloy when solidified under ambient pressure, as shown in Figure 3b (the content of the B2 phase is less and cannot be detected by XRD), but no B2 phase was found after solidification under high pressure. This is because the B2 phase is the result of the ordering of the primary β phase at high temperature [38]; high-pressure solidification can increase the undercooling of the melt and directly lead to the formation of the α phase from the liquid phase [26]. Meanwhile, high pressure inhibits the diffusion which leads to the increasing of Al content in the liquid phase [26]. The content of Al element in the liquid phase will increase, which can also promote the peritectic phase, direct nucleation, and precipitation. Therefore, the B2 phase cannot be found during high-pressure solidification.

Figure 4 shows the EDS analysis result of the Ti-48Al-2Cr alloy solidified under different processes. As shown in Figure 4a, the solubility of Cr element in the lamellar structure is about 2.56 at. % under ambient pressure, whereas it increases to 3.09 at. % and 3.26 at. % under 5 GPa pressure at 1873 K and 1973 K respectively (Figure 4b,c). Considering the errors of EDS, it can be seen that high pressure increases the concentration of Cr in the lamellar structure. Further increasing the holding temperature to 5 GPa, the concentration of Cr is nearly unchanged. Al element in the lamellar structure also shows the same phenomenon. The EDS analysis proves that high pressure inhibits the element diffusion.

Moreover, the EDS results further confirm the decrease of the γ phase and the disappearance of the B2 phase after high pressure solidified.

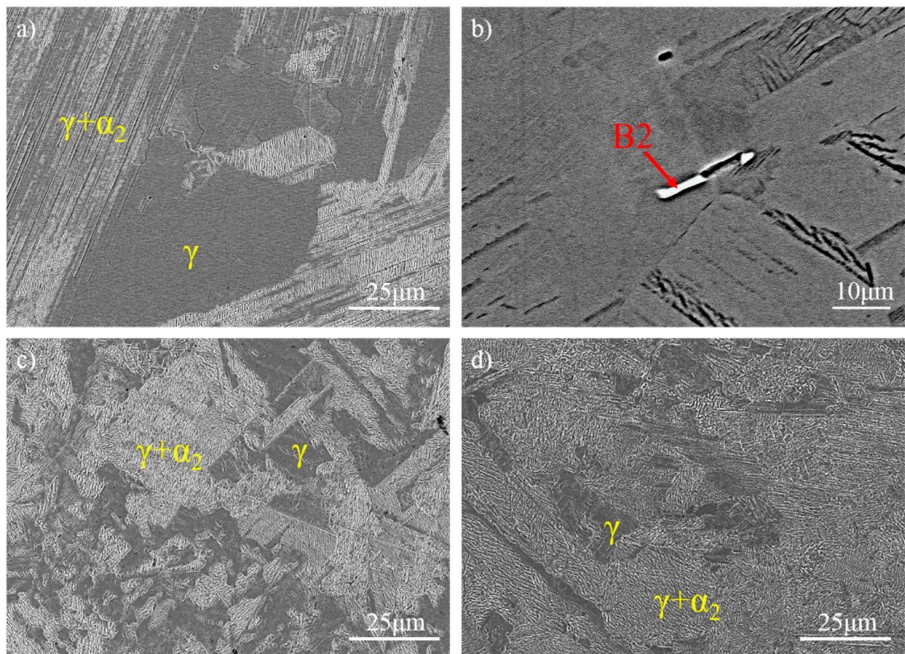

**Figure 3.** Microstructure of the Ti-48Al-2Cr alloy solidified under different temperatures. (**a**) ambient pressure under lower magnification; (**b**) Backscattered Electron (BSE) picture of (**a**) under higher magnification of the B2 phase; (**c**) SEM picture under 5 GPa, 1873 K; (**d**) SEM picture under 5 GPa, 1973 K.

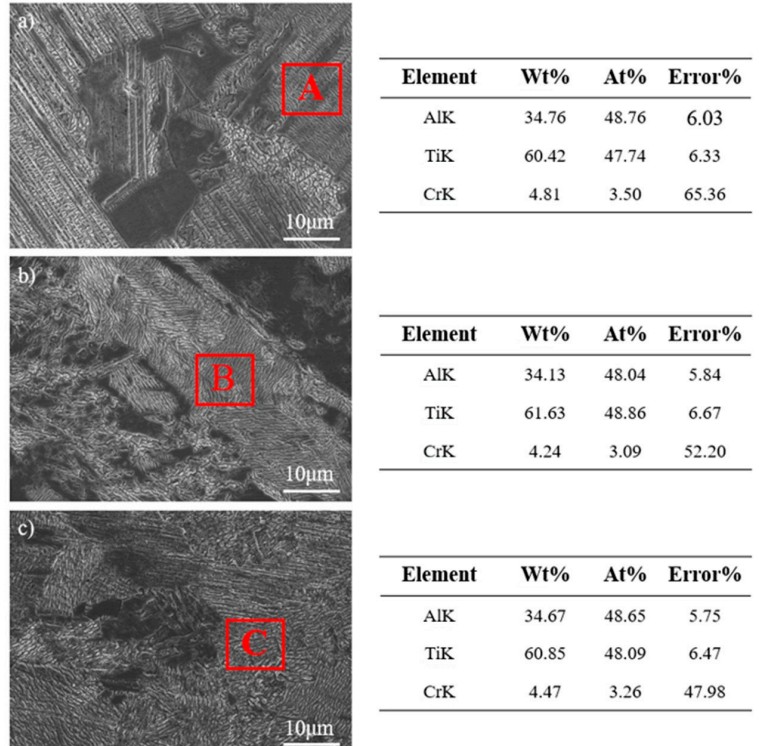

| Element | Wt% | At% | Error% |
|---------|------|------|-------|
| AlK | 34.76 | 48.76 | 6.03 |
| TiK | 60.42 | 47.74 | 6.33 |
| CrK | 4.81 | 3.50 | 65.36 |

| Element | Wt% | At% | Error% |
|---------|------|------|-------|
| AlK | 34.13 | 48.04 | 5.84 |
| TiK | 61.63 | 48.86 | 6.67 |
| CrK | 4.24 | 3.09 | 52.20 |

| Element | Wt% | At% | Error% |
|---------|------|------|-------|
| AlK | 34.67 | 48.65 | 5.75 |
| TiK | 60.85 | 48.09 | 6.47 |
| CrK | 4.47 | 3.26 | 47.98 |

**Figure 4.** EDS analysis of the lamellar structure of the Ti-48Al-2Cr alloy solidified under different pressures. (**a**) ambient pressure, 1873 K; (**b**) 5 GPa, 1873 K; (**c**) 5 GPa, 1973 K.

TEM analysis was carried out to further study the detailed microstructure of the Ti-48Al-2Cr alloy solidified under high pressure. Figure 5 shows several representative areas (SAED) of Ti-48Al-2Cr alloy solidified under 5 GPa at 1873 K and 1973 K. A large volume fraction of dislocations exists in the alloy structure after solidification at 5 GPa and 1873 K (Figure 5a), and the dislocations can be found across the lamellar structure in Figure 5a,b. The number of dislocations in the alloy structure decreases with the increase in holding temperature (Figure 5b,c), indicating that higher holding temperature and longer cooling time lead to the decrease in dislocation density.

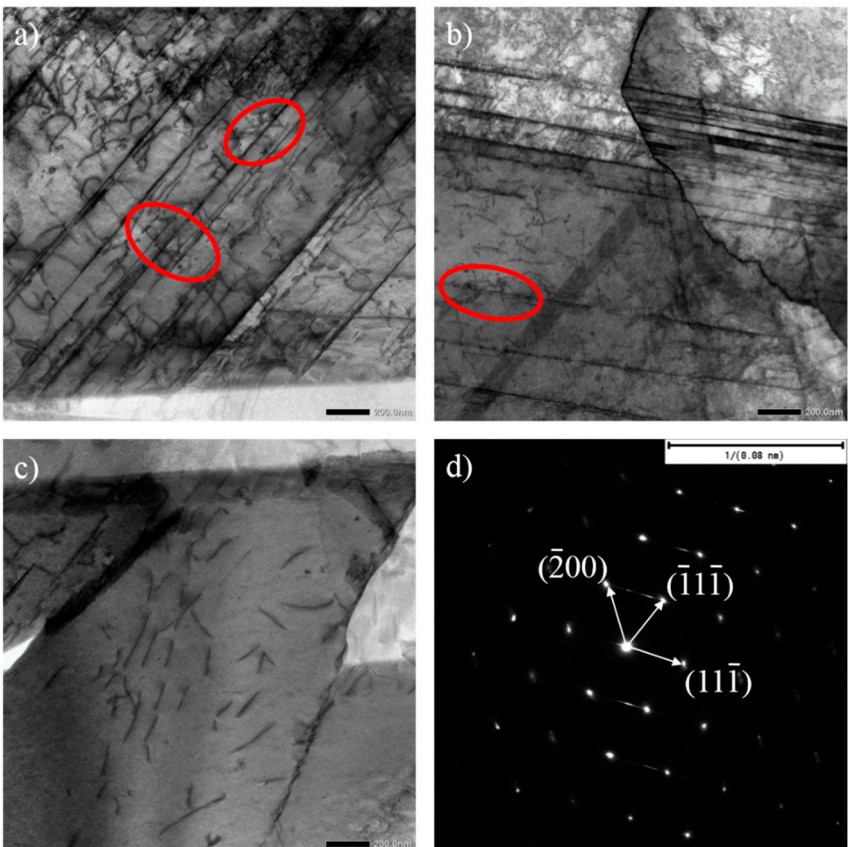

**Figure 5.** TEM images of the Ti-48Al-2Cr alloy solidified at 5 GPa pressure under different temperatures. (**a**) 1873 K; (**b**) and (**c**) 1973 K; (**d**) SAED of γ phase in (**c**).

### 3.3. Nanohardness of High Pressure Solidified Ti-48Al-2Cr Alloy

A nanoindenter was used to measure the mechanical property of high pressure solidified Ti-48Al-2Cr alloy under different processes. Figure 6 shows the hardness-depth curves of the lamellar structures of the Ti-48Al-2Cr alloy solidified under different solidification processes. As can be seen, the fluctuations of the curves are small when the indentation is above 900 nm, which means the hardness of the lamellar structures remains nearly constant with the increase in indentation depth. The average hardness values are displayed in Table 1. When solidified at 5 GPa and 1873 K, the nanohardness is 5.54 GPa. Compared with the samples solidified at ambient pressure, the nanohardness increases about 60.5%. Further increasing the holding temperature led to a decrease in nanohardness.

The nanohardness of the high pressure solidified samples depends on the solidification process. As we mentioned, high pressure leads to an increase in Cr concentration and to dislocations in the microstructures8. Hence, the increase in nanohardness can be attributed to the solution strengthening and dislocation strengthening. When increasing the temperature to 1973 K at 5 GPa, the change of Cr content in the lamellar structures is very small. Combined with the microstructure analysis, the main

reasons for the decrease in nanohardness are the coarsening of the lamellar structure and the decrease in dislocation volume; thus, dislocation strengthening is the dominant mechanism.

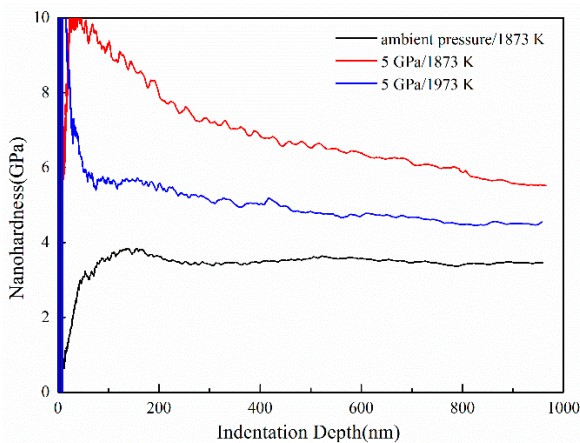

**Figure 6.** Hardness-depth curves obtained from solidified Ti-48Al-2Cr alloy under different pressures.

**Table 1.** The average nanohardness of the Ti-48Al-2Cr alloy solidified under different pressures.

| Sample | Nanohardness (GPa) |
| --- | --- |
| Ambient pressure, 1873 K | $3.45 \pm 0.19$ |
| 5 GPa, 1873 K | $5.54 \pm 0.21$ |
| 5 GPa, 1973 K | $4.48 \pm 0.27$ |

### 3.4. The Mechanism of the Relationship between Dislocation Density, Microstructure Refinement, and Properties under High Pressure

Figure 7 shows a diagram of the dislocation structure relationship of the Ti-48Al-2Cr alloy, which reveals the grain size and dislocation movement in the grain under high pressure and different solidification temperatures. High pressure can inhibit the diffusion of the element and lead to solute enrichment at solidification interface, and supercooling can affect the nucleation and growth of grains and the microstructure evolution mechanism of the alloy [17,22].

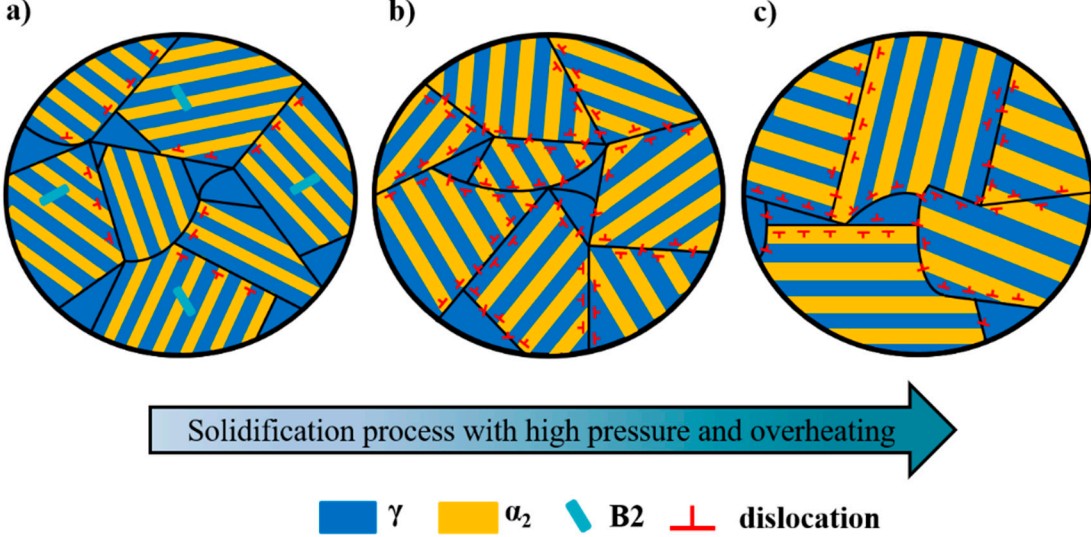

**Figure 7.** The strengthening mechanism in the microstructure of the Ti-48Al-2Cr alloy (**a**) ambient pressure, 1873 K; (**b**) 5 GPa, 1873 K; (**c**) 5 GPa, 1973 K.

Under high pressure, the pile-up of dislocations between grains is relatively close, and the stress produced by high pressure causes the dislocation source to start, release a large number of dislocations, and pile up at the grain boundary (Figure 7a,b). Due to the increased pressure, the dislocations pass through the grain boundary and enter the adjacent grain, which leads to increasing dislocation density and the improvement of mechanical properties [39]. (Figure 5a,b). Compared with the samples solidified at ambient pressure, the nanohardness of the alloy after high-pressure solidification increased to 5.45 GPa and 4.48 GPa, respectively. This is because when solidified at 5 GPa and 1873 K, dislocations cross the grain boundary and enter the adjacent grains, which can activate internal dislocations and release the dislocation pile-ups (Figure 7b). The nanohardness of the alloy was increased by increasing the dislocation density.

The nanohardness of the alloy decreased after being solidified under high pressure at higher superheating temperature. After high pressure solidification, the increased holding temperature led to the increase in dislocation and local stress in the grains. Consequently, the samples after high pressure were more unstable. Therefore, dislocation annihilation plays a dominant role at higher superheating temperatures, which leads to a decrease in dislocation density [40]. It was found that with the increase in solidification temperature, the dislocation density in the $\alpha_2$ phase decreases, and increases in the B2 phase [41]. As was demonstrated in Section 3.2, high pressure led to the increase of the $\alpha_2$ phase and the disappearance of the B2 phase, which further explains the decrease in dislocation density in the samples when solidified under high 5 GPa, 1973 K. (Figure 7c). Therefore, the main strengthening mechanism of the Ti-48Al-2Cr alloy is dislocation strengthening.

## 4. Conclusions

In this paper, the microstructure and mechanical properties of the Ti-48Al-2Cr (at. %) alloys solidified under ambient pressure 5 GPa/1873 K and 5 GPa/1973 K were systematically studied. The conclusions are summarized as follows:

(1) In the process of high-pressure solidification, the volume fraction of interdendritic γ phase decreases while that of the lamellar structure increases.

(2) After high-pressure solidification, the content of Cr and Al increased and the B2 phase disappeared due to the inhibition of element diffusion by high pressure.

(3) The strengthening mechanism of the alloy is solution strengthening and dislocation strengthening after high-pressure solidification. Solidified at 5 GPa and 1873 K, the hardness reached 5.54 GPa. The hardness decreases, and dislocation strengthening plays a dominant role in the superheated state.

**Author Contributions:** Conceptualization, D.Z. and Z.W.; methodology, D.D.; formal analysis, L.L. and D.Z.; investigation, L.L. and Y.L.; resources, D.D., D.Z. and Z.W.; data curation, Y.W. and X.W.; writing—original draft preparation, L.L. and D.Z.; writing—review and editing, D.D. and D.Z. All authors have read and agreed to the published version of the manuscript.

**Funding:** This research was funded by the National Natural Science Foundation of China, grant number 51801112 and 51774105, the Zhejiang Provincial Natural Science Foundation of China grant number LY18E010003.

**Conflicts of Interest:** The authors declare no conflict of interest.

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
