# Peer review of "Microstructure and Nanohardness of Ti-48Al-2Cr Alloy Solidified under High Pressure"

_applsci, doi:10.3390/app10155394_

Round 1
Reviewer 1 Report
In presented manuscript authors described the Ti/Al/Cr alloy solidified under different pressures and temperatures. The effect of high pressure on microstructure and nanohardness of Ti-48Al-2Cr alloy was investigated by powder X-ray diffraction (PXRD), scanning electron microscope (SEM), transmission electron microscopy (TEM) and Nanoindenter XP testing machine.
Some deficiencies exist, which are listed below:
-Method section is not well described for example:
-The authors claimed that: “The phase composition of the samples was determined by X-ray diffraction (XRD)” using “Bruker, D8 Advance” diffractometer. which cannot be true because using Bruker, D8 Advance diffractometer only Powder X-Ray Diffraction PXRD not XRD can be examined.
-The manuscript has few editorial errors that can be corrected at the galley proof stage.
-Figure 4. EDS analysis is of poor quality cannot be read.
-The discussion should be re-written, it should include a literature review and the discussion of the current findings in light of the previous reports in literature. In its current form, the discussion, is not coherent. I believe that there are more than 26 references regarding the microstructure and nanohardness of various materials.
-In the introduction I’m missing reference and discussion of major work on nanohardness studies, no reference to this substantial contribution to this area is given but furthermore, that research should be discussed in these context as it is highly relevant, for example: Appl. Phys. Lett. 2000, 76, 2214; RSC Adv. 2016, 6, 66037; Materials Characterization 2007, 58, 380 but many others are available.
Undoubtedly, the manuscript is very important for technical and technological applications. However, the article is purely technological, scientific novelty (research and discovery of new effects, the study of any physical, chemical and other processes) is missing. However, since this journal is a specialized journal, I believe that the article is suitable for publication after major revision.
Author Response
- Method section is not well described for example:
The authors claimed that: “The phase composition of the samples was determined by X-ray diffraction (XRD)” using “Bruker, D8 Advance” diffractometer. which cannot be true because using Bruker, D8 Advance diffractometer only Powder X-Ray Diffraction PXRD not XRD can be examined.
Reply: We appreciate the Reviewer’s comments. We have corrected the method section in the revised manuscript. It has been revised as: “The phase composition of the samples was determined by X-ray diffraction (XRD) with Cu Kα radiation” (Para.2, Section 2)
- The manuscript has few editorial errors that can be corrected at the galley proof stage.
Reply: The authors are very sorry for the editorial errors in our previous submitted manuscript. The authors re-check the manuscript carefully and make corresponding change in the revised manuscript.
- Figure 4. EDS analysis is of poor quality cannot be read.
Reply: We are very sorry for that the EDS analysis in Figure 4 are of poor quality in our previous submitted manuscript. The authors changed all EDS micrographs and results in Figure 4. (Figure 4, Section 3. 2)
- The discussion should be re-written, it should include a literature review and the discussion of the current findings in light of the previous reports in literature. In its current form, the discussion, is not coherent. I believe that there are more than 26 references regarding the microstructure and nanohardness of various materials.
Reply: We appreciate the reviewer’s comment. The discussion really should include a literature review and the discussion of the current findings in light of the previous reports in literature. The authors checked the introduction carefully and the revisions are listed below:
The sentence “It is because the high pressure shifts the eutectic point to higher Al content and also increases the eutectic reaction temperature. According to the lever rule, the content of lamellar structure increases finally.” has been revised as: “Because of the shifted eutectic point to higher Al content and the increased eutectic reaction temperature, so the content of lamellar structure increases finally according to the lever rule [34-36].” (Para. 2, Section 3. 2)
The sentence “However, increased superheat can coarsen the lamellar structure, in lamellar structures.” has been revised as: “However, according to Gao’s study [37], the effect of temperature on the size and shapes stability of microstructural features has been extensively investigated, the lamellar structure would be coarsen with increasing superheat.” (Para. 2, Section 3. 2)
The sentence “Meanwhile, the diffusion is inhibited greatly under high pressure, which leads to the increasing of Al content in the liquid phase.” has been revised as: “Meanwhile, high pressure inhibits the diffusion which leads to the increasing of Al content in the liquid phase [26].”(Para. 3, Section 3. 2)
The sentence “When the pressure is higher, the dislocations will pass through the grain boundary and enter the adjacent grain boundary, increasing the dislocation density and can further improve the properties of the alloy (Figure 5a-b).” has been revised as: “By the increased pressure, the dislocations passes through the grain boundary and enters the adjacent grain which lead to the increasing of dislocation density and the improving of mechanical properties [39].” (Para. 2, Section 3. 4)
The author added references and checked the format of the references carefully.
- In the introduction I’m missing reference and discussion of major work on nanohardness studies, no reference to this substantial contribution to this area is given but furthermore, that research should be discussed in these context as it is highly relevant, for example: Appl. Phys. Lett. 2000, 76, 2214; RSC Adv. 2016, 6, 66037; Materials Characterization 2007, 58, 380 but many others are available.
Reply: We thank the reviewer for his comments. The following sentence has been added in the revised manuscript. “Nanoindentation is a test technique which has widespread for measuring the mechanical properties of the nanoscale, the microscale and the thin-films materials [28-29]. It makes it successful to determine the hardness of the material especially the hardness of target phase from hardness-depth data [29-30]”. (Para. 2, Section 1)

Reviewer 2 Report
In the paper entitled “Microstructure and Nanohardness of Ti-48Al-2Cr Alloy Solidified under High Pressure”, Authors present the results of influence of pressure on the solidification of the examined alloy. Presentation results is clear and contains interesting information. Article is very interesing.
The paper could be published after the following comments are addressed. Detailed comments are given below:
1) The introduction does not include a specific application of the material.
2) The phrase "chromium alloy (99.99%)" is not correct. It's not a alloy.
3) What thermocouple was used?
4). Please improve the readability of Figure 4 (EDS). The error of EDS analysis should be given (fig.4a,b,c)
After considering the suggestions, the article is printable.
Author Response
In the paper entitled “Microstructure and Nanohardness of Ti-48Al-2Cr Alloy Solidified under High Pressure”, Authors present the results of influence of pressure on the solidification of the examined alloy. Presentation results is clear and contains interesting information. Article is very interesing.
The paper could be published after the following comments are addressed. Detailed comments are given below:
- The introduction does not include a specific application of the material.
Reply: Thank you for your comments. Due to the low density, high oxidation resistance and high-temperature strength of TiAl alloys, it can be used as structural materials at high temperature. Alloying was used to improve the mechanical property of TiAl alloy. The corresponding revisions have been added in the revised manuscript. “Alloying elements such as Cr, Nb and Zr were added to improve the room temperature brittleness of the TiAl based alloy [31]. It has been considered that 2% (at.%) Cr additions add to Ti-48Al alloy have the best room temperature ductility [13]. ” (Para. 3, Section 1)
- The phrase "" is not correct. It's not a alloy.
Reply: It is really true as reviewer pointed out: “…chromium alloy (99.99%) …” should be corrected as “…high purity chromium (99.99%) …” in the description in ‘Materials and Methods’ part. (Para. 1, Section 2)
- What thermocouple was used?
Reply: The W3%Re-W-25%Re thermocouple (type D) thermocouple was used in this study. The authors have added the experimental details in the revised manuscript. (Para. 1, Section 2)
- Please improve the readability of Figure 4 (EDS). The error of EDS analysis should be given (fig.4a,b,c)
Reply:We are very sorry for the EDS results readability, again. The authors revised the EDS micrographs and added the error of EDS analysis in the corresponding figure. (Figure 4, Section 3.2)

Reviewer 3 Report
In the manuscript titled “Microstructure and Nanohardness of Ti-48Al-2Cr Alloy Solidified under High Pressure”, the authors present the improvement of mechanical properties in TiAl based alloys by using high pressure and high temperature technique. The entire study is interesting with some reasonable discoveries. However, before the final decision (accept or reject), the following concerns should be addressed by the authors.
- High pressure and high temperature technique is an important and robust approach widely used in basic and applied science as well as industrial applications. So, I highly recommend to include a few sentences in the manuscript to highlight this fact. Here several papers (Journal of superhard materials 34 (6), 360-370, 2012; Scientific Reports volume 2, Article number: 520 (2012); Applied Physics Letters 108 (6), 061906, 2016; Physical Review B 92 (6), 064108, 2015)) are provided for the reference purpose.
- The pressure was increased to 5 GPa during the experimental process, so how the pressure was determined?
- In sentences 106 to 107, the phase diagram of Ti-Al-Cr alloy was mentioned. Where this phase diagram comes from? If it comes from published materials, please add the references by using a proper way. Also, please briefly explain the meanings of the letters, such as L, α, β, and so on.
- High temperatures would dense the material, which may increase the hardness of the material, but the study shows a reverse trend. Any explanation?

Author Response
In the manuscript titled “Microstructure and Nanohardness of Ti-48Al-2Cr Alloy Solidified under High Pressure”, the authors present the improvement of mechanical properties in TiAl based alloys by using high pressure and high temperature technique. The entire study is interesting with some reasonable discoveries. However, before the final decision (accept or reject), the following concerns should be addressed by the authors.
- High pressure and high temperature technique is an important and robust approach widely used in basic and applied science as well as industrial applications. So, I highly recommend to include a few sentences in the manuscript to highlight this fact. Here several papers (Journal of superhard materials 34 (6), 360-370, 2012; Scientific Reports volume 2, Article number: 520 (2012); Applied Physics Letters 108 (6), 061906, 2016; Physical Review B 92 (6), 064108, 2015)) are provided for the reference purpose.
Reply:It is really true as reviewer suggested that the authors should refer more references to highlight the importance of high pressure and high temperature technique. The following revision has been made to improve the quality of this paper. “High pressures and high temperatures have revolutionary impacts in the nowadays society ranging from industrial applications such as the fabricate of superconducting and superhard materials, to exploration the properties of materials obtained under extreme conditions [22-25]”. (Para. 2, Section 1)
- The pressure was increased to 5 GPa during the experimental process, so how the pressure was determined?
Reply:We thank the reviewer for his comments. The phase composition of Bi varies with resistance, the circuit in a high-pressure cubic cell can determine the changes of Bi’s resistance, so the pressure at different positions can be measured. The detailed pressure measurement of was described in previous studies [32]. (Para. 1, Section 2)
- In sentences 106 to 107, the phase diagram of Ti-Al-Cr alloy was mentioned. Where this phase diagram comes from? If it comes from published materials, please add the references by using a proper way. Also, please briefly explain the meanings of the letters, such as L, α, β, and so on.
Reply:It is really true as reviewer suggested that the references of Ti-Al-Cr alloy phase diagram [32] should be added. We have explained the meanings of the letters, such as L, α, β, and so on. (Para. 1, Section 3.2)
- High temperatures would dense the material, which may increase the hardness of the material, but the study shows a reverse trend. Any explanation?
Reply:We thank the reviewer for his comments. In this study, the nanohardness of the Ti-48Al-2Cr alloy is 3.45 GPa solidified at ambient pressure. When solidified at 5 GPa/1873 K, the hardness raises to 5.54 GPa which increases by 60.5% compared with that solidified at ambient pressure. Compared with the ambient pressure sample, the nanohardness increased by 29.9% solidified at 5 GPa/1973 K. So, the hardness increases with the increasing temperature. At 5 GPa, the increasing of superheating would decrease the hardness and the corresponding reason has been clarified in section 3.4.

Round 2
Reviewer 1 Report
The revised version of the manuscript could be accepted for publication. Authors have worked hard on their paper.
Congratulations to the authors.
